# Investigation of Effects of Novel *Bifidobacterium longum* ssp. *longum* on Gastrointestinal Microbiota and Blood Serum Parameters in a Conventional Mouse Model

**DOI:** 10.3390/microorganisms12040840

**Published:** 2024-04-22

**Authors:** Merle Rätsep, Kalle Kilk, Mihkel Zilmer, Sirje Kuusik, Liina Kuus, Mirjam Vallas, Oksana Gerulis, Jelena Štšepetova, Aivar Orav, Epp Songisepp

**Affiliations:** 1BioCC OÜ, Riia St. 181A, 50411 Tartu, Estonialiina.kuus@biocc.ee (L.K.); mirjam.vallas@biocc.ee (M.V.);; 2Department of Biochemistry, Institute of Biomedicine and Translational Medicine, University of Tartu, Ravila St. 19, 50411 Tartu, Estonia; kalle.kilk@ut.ee (K.K.);; 3Department of Microbiology, Institute of Biomedicine and Translational Medicine, University of Tartu, Ravila St. 19, 50411 Tartu, Estonia; 4Tartu Health Care College, Nooruse St. 5, 50411 Tartu, Estonia

**Keywords:** *Bifidobacterium longum*, mouse model, probiotic, microbiota, metabolomics, health parameters

## Abstract

Representatives of the genus *Bifidobacterium* are widely used as probiotics to modulate the gut microbiome and alleviate various health conditions. The action mechanisms of probiotics rely on their direct effect on the gut microbiota and the local and systemic effect of its metabolites. The main purpose of this animal experiment was to assess the biosafety of the *Bifidobacterium longum* strain BIOCC1719. Additional aims were to characterise the influence of the strain on the intestinal microbiota and the effect on several health parameters of the host during 15- and 30-day oral administration of the strain to mice. The strain altered the gut microbial community, thereby altering luminal short-chain fatty acid metabolism, resulting in a shift in the proportions of acetic, butyric, and propionic acids in the faeces and serum of the test group mice. Targeted metabolic profiling of serum revealed the possible ability of the strain to positively affect the hosts’ amino acids and bile acids metabolism, as the cholic acid, deoxycholic acid, aspartate, and glutamate concentration were significantly higher in the test group. The tendency to increase anti-inflammatory polyamines (spermidine, putrescine) and neuroprotective 3-indolepropionic acid metabolism and to lower uremic toxins (P-cresol-SO_4_, indoxyl-SO_4_) was registered. Thus, *B. longum* BIOCC1719 may exert health-promoting effects on the host through modulation of the gut microbiome and the host metabolome via inducing the production of health-promoting bioactive compounds. The health effects of the strain need to be confirmed in clinical trials with human volunteers.

## 1. Introduction

Representatives of *Bifidobacteria* are normal commensals of the human gastrointestinal microbiota that colonise the host’s gut throughout their lifespan. Bifidobacteria quickly colonise infant intestines during the first weeks of life and reach the highest proportion in the colon during the first 12 months after birth. Representatives of *Bifidobacteria* are the first inhabitants in the gut of infants due to their complex genetic pathways to metabolise different glycans present in human milk. *Bifidobacteria* help to maintain intestinal microbial balance and health by contributing to the anaerobic environment in the gut and lowering the pH, making it less favourable for potential pathogenic bacteria, etc., [1].

Various oligosaccharides (human milk oligosaccharides, HMOs) in human breast milk influence the colonisation of the infant gut with different *Bifidobacterium* species and increase their species diversity as well as the diversity of the entire microbiota through cross-feeding and environmental modification.

In adults, dietary fibres mimic the effect of HMOs. Dietary fibres are one of the factors that affect the diversity of *Bifidobacteria* in the adult gut due to their genetic diversity that helps them to utilise different types of fibre, including non-starch polysaccharides, cellulose, pectin, hydrocolloids, fructo-oligosaccharides, and resistant starch [2,3].

Along with *Lactobacillus* spp., different *Bifidobacterium* species are historically associated with food and have also been widely used as probiotics to modulate the gut microbiome and alleviate various conditions [4,5].

Probiotics are generally considered safe for a healthy population. Adopted by the FAO and [6] WHO in 2021, the probiotic definition of “live microorganisms that, when administered in adequate amounts, confer a health benefit on the host” has been retained ever since [7]. However, different consumers may have very different perceptions of “adequate amounts”, which may result in side effects such as bloating or accelerated digestion. In addition, some adverse effects of probiotics, including *Bifidobacterium*-based probiotics, have been reported, such as systemic infections and deleterious metabolic activities. These reports also cover cases of bacteraemia linked to the species *B. longum*, which are related primarily to vulnerable population groups (e.g., pre-term infants or other population groups with a weakened immune system) [8,9,10].

Therefore, just as the definition of probiotics has not become obsolete, the international recommendations for the stepwise development of any novel microbial strain into a probiotic have not become useless either [6]. Even strains from historically “safe” species from human samples should be scrutinised.

The mechanisms of the action of probiotics depend on their direct effect on the digestive tract’s microbiota and the effect of its metabolic end products, which can be both local and systemic. Unravelling the mechanisms underlying the potential probiotic effects of a microbial strain requires evaluating three different levels of its properties: the widespread mechanisms associated with a particular taxonomic group, more frequent species-specific effects, and, finally, rarer strain-specific effects [7].

Recently, we have reported a novel *Bifidobacterium longum* ssp. *longum* strain BIOCC 1719 (BL1719) of healthy child origin [11]. We characterised the metabolic diversity of the strain by determining bioactive compounds like B vitamins, essential amino acids, and fatty acids that the strain simultaneously produced in different growth media during in vitro cultivation. We aimed to exploit the properties of this strain to develop a postbiotic that can enrich human nutrition.

The aim of this study was to assess the biosafety of BL1719 on a mouse model and to elucidate its potential impact on the gut microbiota and the metabolic profile of the host when administered in live form at a specific dosage. 

## 2. Materials and Methods

### 2.1. Origin of the Strain

The microorganism *Bifidobacterium longum* ssp. *longum* strain BIOCC 1719 (DSM 34239) (hereinafter referred to as BL1719) was isolated from a stool sample of a 2-month-old full-term breastfed Estonian child in 2020. The strain-specific bioactive compounds produced during in vitro cultivation and the effect of the growth media have been characterised previously [11].

### 2.2. Animal Trial

#### 2.2.1. Experimental Design

The trial protocol was approved by the Ethics Committee on Animal Experiments of the Ministry of Regional Affairs and Agriculture of Estonia (reference number 215/11.04.2022). Conventional male mice were used in this study (*n* = 32 balb/c; the Laboratory Animal Centre, Institute of Biomedicine and Translational Medicine, Faculty of Medicine, University of Tartu, Estonia). At the beginning of the experiment, the median (quartiles) age of the mice was 89 (87;90) days. 

The experiment consisted of two intervention periods: a short intervention (15 days) and a long intervention (30 days) (Figure 1). The mice came from three different litters and were randomly divided into two groups: a test group receiving BL1719 (TG) and a control group (CG). According to the duration of the intervention, both groups were further divided into two subgroups, resulting in total four groups of mice: TG1, TG2, CG1, and CG2, each with eight animals per cage (Figure 1).

All the mice were housed in ventilated cages bedded with aspen bedding (Millamore Supesoft, Tapvei Estonia OÜ, Estonia) and were fed with a commercial diet (rat/mice universal maintenance diet V1534, ssniff Spezialdiäten GmbH, Soest, Germany) and (distilled) water ad libitum. BL1719 was administered as a suspension in the drinking water (10 log_10_ cfu/mL) from a 24-hour-old culture. The water bottles were changed daily, and the amount of consumed water was measured. The daily dose of BL1719 per mouse per day was calculated based on average of the amount of consumed water per animal in the cages. The control animals received distilled water. The health and well-being of the mice were monitored daily. Body weight was measured weekly.

Half of the mice of both groups (TG and CG) were sacrificed by decapitation on day 15 of the experiment (i.e., subgroups TG1 and CG1), and the remaining 8 mice of both groups were sacrificed on day 30 (i.e., subgroups TG2 and CG2) (Figure 1).

#### 2.2.2. Sample Collection 

Faeces samples were collected per cage for three consecutive days before the intervention and for three consecutive days before the end of the intervention (days 13, 14, and 15 for the short intervention, and days 28, 29, and 30 for the long intervention) (Figure 1). 

Tissue samples for histological analysis were collected accordingly at the end of the intervention period on day 15 (TG1 and CG1) and on day 30 (TG2 and CG2). Tissue samples of the liver, spleen, small intestine, and large intestine were collected, fixed in 10% formaldehyde, and embedded in paraffin. The microtome-cut tissue samples were stained with hematoxylin and eosine. 

Blood samples were collected accordingly in aseptic conditions at the end of the intervention period on day 15 (TG1 and CG1) and on day 30 (TG2 and CG2). The samples were collected for metabolic profiling and to identifying the possible translocation of gut microbiota and BL1719.

#### 2.2.3. Sample Analysis

##### Translocation Analysis

Volumes of 100 µL of blood were spread on TPY agar (tryptone and papainic digest of soya, Condalab, Laboratorios Conda S.a., Madrid, Spain) to detect bifidobacterial translocation and on sheep blood agar (Mikrolabor OÜ, Tallinn, Estonia) to detect the translocation of other gastrointestinal microbes. After anaerobic incubation for 48 h at 37 °C (H_2_ 5%/CO_2_ 5%/N 90% (product code 100390, AS Linde Gas, Tallinn, Estonia; anaerobic workstation “Concept 500”, Baker Ruskinn, Bridgend, South Wales)), the appearance of any microbial colonies was considered a translocation.

##### DNA Extraction from Faecal Samples

To extract DNA from the stool samples, all the collected faecal samples per cage (i.e., per group) were finely ground. The faecal samples were homogenised mechanically. DNA from 0.2 g of the homogenised samples was extracted according to the QIAamp DNA Stool Kit protocol (QIAGEN, 2010[ES1]). DNA 16S amplicon sequencing was performed by Novogene Co., Ltd. (Cambridge, UK). The V3–V4 region was amplified using the *341F* (5-*CCTAYGGGRBGCASCAG*-3) and *806R* (5-*GGACTACNNGGGTATCTAAT*-3) primers [12]. Sequences were obtained on an Illumina MiSeqTM platform in a 2 × 300 bp paired-end run (Illumina, Inc. San Diego, CA, USA) following the standard instructions of the 16S Metagenomic Sequencing Library Preparation protocol. The gut microbiome sequencing data were processed in the Qiime2 software (version 2021.4) [13].

#### 2.2.4. Analysis of Short-Chain Fatty Acids from Faecal Samples

Short-chain fatty acids (SCFAs) from the faecal samples were extracted according to a modified procedure of Zhao et al. [14]. SCFAs were determined by an Agilent 6890A (Agilent Technologies, Inc., 5301 Stevens Creek Blvd. Santa Clara, CA USA) gas chromatograph, a CP-Wax 52 CB capillary column (30 m × 0.25 mm, 0.25 μm), and a flame ionisation detector at 280 °C. The oven temperature program was as follows: hold at 175 °C for 1 min, increase by 20 °C/min to 190 °C, and then hold for 5 min. Acetic acid, propionic acid, butyric acid, isobutyric acid, isovaleric acid, valeric acid, and hexanoic acid solutions were used for identification and calibration. The results were expressed as µmol/g sample. Time point measurements of the SCFA content or microbial abundancies per group were calculated as medians from three SCFA content measurements or microbial abundancies in consecutive days before the time point.

#### 2.2.5. Analysis of Short-Chain Fatty Acids from Blood Serum

SCFAs were derivatised with 2-nitrophenylhydrazine by treating each 50 μL sample with 50 μL of internal standards (^2^H_4_acetic acid and ^2^H_11_hexanoic acid), 100 μL of 200 mmol/l 2-nitrophenylhydrazine, and 20 μL of 120 mmol/L 1-(3-dimethylaminopropyl)-3-ethylcarbodiimide. After 1 h of incubation, the mixture was centrifuged for 10 min at 21,000× *g*. The SCFAs were separated on an ACQUITY Premier BEH C18 column with a VanGuard FIT, 1.7 µm, 2.1 × 100 mm^2^ column using water and acetonitrile with 0.1% formic acid eluents. The results were expressed as µmol/L. 

#### 2.2.6. Targeted Metabolic Profiling of Blood Serum

Targeted metabolic profiling of the blood serum of the mice was performed with a Xevo TQ-XS mass spectrometer (Waters, MA, USA) coupled with an ACQUITY ultra-performance liquid chromatography device (Waters, MA, USA) using an MxP^®^ Quant 500 kit (Biocrates Life Sciences AG, Vienna, Austria, https://biocrates.com). The samples were prepared and analysed according to the manufacturer’s instructions, and the results were expressed as µmol/L.

#### 2.2.7. Lipid Level Measurement in Mouse Blood Serum

Total cholesterol, high-density lipoprotein cholesterol (HDL cholesterol), low-density lipoprotein cholesterol (LDL cholesterol), and triglycerides from the blood serum of the mice were analysed with a Clinical Chemistry Analyzer Mindray 230 device (Mindray Bio-Medical Electronics Co., Ltd., Shenzhen, China). The samples were manually pre-diluted with saline as tenfold dilutions. Total cholesterol and triglycerides were analysed with an enzymatic photometric method (product code for cholesterol kit: 0045, and product code for triglycerides kit: 0075, Giesse diagnostic, Guidonia Montecelio, Italy) according to the manufacturer’s instructions, and HDL cholesterol and LDL cholesterol were analysed with a direct enzymatic method (product code for HDL cholesterol kit: 105-00083500, and product code for LDL cholesterol kit: 105-000836-00, Mindray Bio-Medical Electronics Co., Ltd., Shenzhen, China) according to the manufacturer’s instructions.

#### 2.2.8. Statistical Analyses

Statistical analyses were performed using the R software, version 4.3.0 (The R Foundation for Statistical Computing, Vienna, Austria), and MS Excel. Longitudinal analysis of the weekly change in body weight over the 4-week study period was performed by a linear mixed model, including the fixed effects of week, groups with their interaction, and random effects of cage and mouse. Blood serum markers were compared between the study groups by Wilcoxon’s rank sum test. Statistical significance was declared at *p* < 0.05. Analysis of the beta-diversity of studied groups, including the PCoA of weighted UniFrac distances, were performed and visualised using PAST version 4.0 [15].

## 3. Results

The main purpose of this animal experiment was to assess the biosafety of the novel *Bifidobacterium* strain BL1719. Additionally, we aimed to obtain as broad preliminary information as possible about the possible interaction(s) of the strain with the host. We hereby report the local as well as the systemic effects of the oral administration of this strain on several health parameters.

### 3.1. Safety Assessment

An average of 4.5 mL of drinking water was consumed per mouse for 24 h. As the strain BL1719 was administered as a suspension in the drinking water (10 log_10_ cfu/mL), the daily dose of BL1719 was 10.65 log_10_ cfu/mouse.

During the experiment, all the mice stayed in good condition, and no changes in behaviour, the characteristics of the fur, or digestion were detected. No translocation of bacteria to blood was detected. The liver, spleen, small intestine, and large intestine samples obtained at autopsy were sterile in all the mice. No pathological shifts and no micro abscesses, granulomas, or inflammation were found by morphological and histological evaluation of the spleen, liver, ileum, and colon of the mice.

### 3.2. Body Weight

At the start of the intervention, there were no statistically significant differences in body weight between the study groups (*p* = 0.401). The mean body weight in group CG was 28.1 g (SD = 1.65), whereas the mean in group TG was 27.5 g (SD = 2.12). However, there was a statistically significant difference in the body weight change between the study groups over 30 days (*p* < 0.001). According to the estimated model, the body weight gain during the 4 weeks in group CG was 0.19 g/week, while the corresponding estimate in group TG was 0.65 g/week.

### 3.3. Faecal Microbiota

While the primary objective of this animal experiment was centred around the biosafety testing of BL1719 using conventional mice with natural microbiota, it is important to acknowledge the significant role that the gut microbiota plays in host health. Considering that the primary target of probiotic action is the intestinal tract, one of our additional aims was to characterise the influence of BL1719 administration on the composition of the microbiota.

Illumina sequencing of the V3-V4 region of 16S *r*RNA was applied to study the mouse faecal microbiota. A total of 262,912 high-quality reads were obtained, with 6391 ± 5279 reads per cage. OTUs were classified into known taxa, including 27 phyla, 60 genera, and unclassified groups.

Initially, in both cages of the l subgroups (TG1 and TG2), the number of species (richness) was similar, while the Shannon diversity index, H, was higher in the TG2 group (Table 1). After the intervention period, the richness and Shannon diversity index H increased non-significantly in both the TG1 and TG2 groups. In the control cages (CG1 and CG2), the richness and Shannon diversity index, H, was higher at the beginning of the intervention period compared to the TG1 and TG2 groups, but both decreased non-significantly by the end of the intervention.

A principal coordinate analysis (PCoA) plot at the genera level was constructed to assess the relationships between the community structures of the studied samples. The faecal microbiota collected from the different cages changed differently, especially after a longer intervention period, indicating that the microbial communities were structurally different from each other (Figure 2).

*Bacillota*, *Bacteroidota,* and *Candidatus Patescibacteria* were the most abundant phyla in all the samples (Figure 3a). The relative abundance of *Bacillota* was highest in both the CG2 and TG2 groups at the start and decreased during the 30-day period of the intervention (69.3% vs. 22.8% in CG2 and 55.2% vs. 47.7% in TG2, respectively). On the other hand, the relative abundance of *Bacillota* in the CG1 and TG1 groups at the start was lower and increased after 15 days of intervention (38.4% vs. 49.6% in CG1 and 29.9% vs. 42.5% in TG1, respectively). The relative abundance of *Bacteroidota* increased in the CG2 and TG2 groups and decreased in the CG1 and TG1 groups. The relative abundance of *Candidatus Patescibacteria* increased in all the study groups.

Additionally, by the end of the intervention, the relative abundance of the phylum *Desulfobacterota* was 2-fold less in the CG2 group (1.6% vs. 0.73%) and almost 3-fold less in the TG2 group (1.46% vs. 0.55%).

The relative abundance of *Actinobacteriota* increased in the TG group after the administration of the strain BL1719 for both the 15- and 30-day interventions (Figure 3a).

At the genera level, the most abundant genera in all the samples were the *Muribaculaceae*, *Bacteroides*, *Alistipes*, *Clostridia* UCG-14, and *Lacnospiraceae* NK3A136 groups (Figure 3b).

At the genera level, the abundance of specific members of the *Bifidobacterium*, *Lactobacillus*, and *Eubactrium xylanophilum* groups increased in the test group, while an increase in the abundance of *Bacteroides* was observed only in the control group (Appendix A). The relative abundance of *Roseburia* increased in both groups, and *Candidatus Saccharimonas* increased in both the studied groups.

One limitation of the study was the inability to detect the survival of the strain during transit through the gastrointestinal tract. This may have been contributed to by the fact that the process of homogenisation of the faecal samples negatively affected the survival of the administered strain. 

Although the genus Bifidobacterium was represented in the gut microbiota of the mice in both groups (CG and TG), unlike the experimental group, it did not enter the top 20 genera in the microbiota of the control group mice in either intervention period.

### 3.4. Short-Chain Organic Acids in Faeces and Serum

Initially, the SCFA contents in the faecal samples of the TG1 and CG1 groups were lower than in the TG2 and CG2 groups (Table 2). The administration of the BL1719 strain for 15 days and 30 days resulted in a higher content of total faecal SCFAs and total straight-chain SCFAs in the TG1 and TG2 groups in comparison with the respective controls (CG1 and CG2).

The dynamics of SCFAs during the longer feeding period in the TG2 and CG2 groups were different. In the TG2 group, a steady increase in total SCFAs and total straight-chain-SCFAs was recorded, while in the CG2 group, both parameters increased during the first 15 days and then decreased during the following 15-day period to approximately the same level as at the beginning (Table 2).

The total BCFA content in the faecal samples at the end of the 15-day feeding period was also somewhat higher in the TG1 group than in the CG1 group but reached the same level in the TG2 and CG2 groups.

During the short intervention, the dynamics in the faecal acetic acid and butyric acid concentrations in the CG1 and TG1 groups were similar. At the end of the 15-day feeding period, higher acetic acid (68.11 µmol/g) and lower butyric acid (8.75 µmol/g) contents were recorded in the TG1 group in comparison with the respective values in the CG1 group (61.32 µmol/g for acetic acid; 9.83 µmol/g for butyric acid) (Table 3). However, the change in the acetic acid concentration from the start was higher in the TG group (1.33 times) in comparison with the value of 1.03 times in the CG1 group. The increase in the butyric acid concentration was similar in both groups of mice. The dynamics of the propionic acid concentration during the short intervention were opposite for the TG1 and CG1 groups, i.e., an increase was observed in the TG1 group, and a decrease was observed in the CG1 group (Table 3).

In the TG2 group, the dynamics of the faecal acetic acid and propionic acid concentrations were similar in both the 15-day periods (i.e., the short intervention and the first 15 days of the long intervention) (Table 3 and Table 4). During the whole 30-day intervention period, a steady increase in faecal acetic acid and propionic acid was recorded in the TG2 group (Table 4). After the 30-day administration of BL1719, the faecal acetate level remained higher in the TG2 group (74.30 µmol/g vs. 69.33 µmol/g in CG2). Most anaerobes, including the genera *Bifidobacterium* and *Ruminococcus*, produce acetic acid, an essential substrate for butyrate-producing colon bacteria. Acetic acid decreases the luminal pH in the colon, has anti-inflammatory effects, and is used by colonic bacteria to produce butyric acid [16].

The butyric acid content in the faeces decreased in the TG2 group during the first 15 days in the longer feeding period in contrast to the shorter intervention, and the decline continued the following 15 feeding days. Despite the decrease during the longer intervention, the butyric acid level (8.49 µmol/g) in the TG2 group still exceeded the corresponding value (8.21 µmol/g) in the CG2 group on day 30 (Table 4). In the intestinal tract, butyric acid is the preferred energy source for enterocytes and colonocytes, helping to maintain the health of epithelial cells, improving gut barrier function, stimulating the proliferation of normal colonic epithelial cells, and possessing anti-inflammatory effects, which protect against colon colitis and cancer, etc. [17].

Like other SCFAs, butyric acid also acts as a signalling molecule between the microbiota and host. For example, Onyszkiewicz and co-workers demonstrated in a study with rats that butyric acid produced by the gastrointestinal microbiota exerts a systemic effect via the gut–brain axis, acting as a hypothetical agent over colon–vagus nerve signalling [18].

Although this animal experiment is still a model system and one should be careful with the interpretation of the findings, the more erratic dynamics of the three main SCFAs in the CG group in comparison with the TG group hints that administration of BL1719 helps to maintain the gastrointestinal microbiota community.

Lactic acid is another important substrate for butyrate-producing bacteria. Lactic acid is produced by lactic acid bacteria, including *Lactobacillus* spp., and, to some extent, bifidobacteria. Although the abundance of the genus *Lactobacillus* increased in the TG1 and TG2 groups during both the short and long intervention periods, the lactic acid contents remained below the detection level in the faecal samples. One possibility is that the formed lactic acid was metabolised by lactate-utilising species to produce the short-chain fatty acids butyrate and propionate [19]. However, a more likely explanation is methodology-based. In the faeces, the fatty acid profile was determined by gas chromatography, while in the blood serum, a commercial kit and an HPLC mass spectrometer were used.

SCFAs produced by colonic bacteria have several physiological effects on the human body. Through the portal vein, acetic acid serves as a substrate for cholesterol and fatty acid biosynthesis in the liver and is used by muscle and brain tissue as an energy source [16].

The serum acetic acid content reached approximately the same levels in the TC and CG groups during both the short and long intervention periods (Table 5). Although the propionic acid concentration in the TG group was somewhat higher on day 15 and lower on day 30, the difference was not significant.

After 15 days of BL1719 consumption, the serum butyric acid content remained lower in the TG1 group (0.46 µmol/L vs. 0.86 µmol/L in CG1, *p* = 0.09) and significantly lower (*p* = 0.016) in the TG2 group (0.63 µmol/L) in comparison with the CG2 group (1.06 µmol/L) at the end of 30-day feeding period (Table 5). This may indirectly indicate a better health status of the colonocytes and thus the integrity of the intestinal barrier, since butyric acid is the main energy source of colonocytes [20]. Both the serum lactic acid and succinic acid contents in the TG group remained lower than the respective values in the CG group at the end of both intervention periods (Table 5).

Three straight-chain SCFAs (acetic, butyric, and propionic acids) formed the majority of both the faecal and serum SCFAs (Figure 4). In comparison with the other SCFAs, acetic acid was proportionally the highest in both groups of mice (TC and CG) and higher in the serum samples (over 80% of the total SCFAs) compared to the faecal samples, where it remained below 80% during both exposure periods. Propionic acid and butyric acid followed.

When comparing the two intervention periods, the proportions of propionic acid and butyric acid in the faeces were opposite. At the end of the shorter intervention, the proportion of propionic acid in both groups was 9%, and that of butyric acid was 12% in the CG1 group and 10% in the TG1 group. At the end of the longer intervention, however, the opposite was true; the proportion of propionic acid was 12% in the CG2 group and 10% in the TG2 group, while the propionic acid proportion remained 9% in both groups.

The acetic acid content was somewhat higher in the TG faecal samples after both intervention periods (79%) in comparison with a content of 77% in the CG faecal samples. 

In the serum samples, a higher proportion of acetic acid among the SFCAs became evident in the TG group after the longer exposure (87% in TG2 vs. 84% in CG2) (Figure 4).

At the expense of a lower acetic acid content, butyric acid was proportionally higher in the faecal samples in the CG1 group (12% vs. 10% in TG1) and in the serum samples at the end of the longer intervention period (6% in CG2 vs. 3% in TG2) (Figure 4).

### 3.5. Targeted Metabolic Profiling of Blood Serum

Bile acids are involved in lipid metabolism, including cholesterol biosynthesis and elimination, the promotion of lipid- and fat-soluble vitamin absorption, etc. [21].

Primary bile acids are synthesised de novo by the liver from cholesterol. Secondary bile acids are produced by intestinal microbes from primary bile acids, which are excreted into the intestinal tract as glycine or taurine conjugates [21].

After 15 and 30 days of BL1719 administration, the serum levels of cholic acid (CA, one of the primary bile acids) and deoxycholic acid (DCA, one of the secondary bile acids) were observed to increase in the TG1 and TG2 mice, although the difference from the respective values in the CG group was significant only at the end of the shorter intervention period (Table 6).

An increase in DCA is associated with the microbial bile salt hydrolase activity of intestinal bacteria, which in turn is one of the main mechanisms of the hypocholesterolaemic effect due to the more efficient removal of bile acids from the body [22]. Jain et al. have previously described the importance of DCA in colonic wound repair. DCA regulates prostaglandin E2 (PGE2) levels via the FXR receptor, and it thus plays a significant role in guiding the different phases of the wound repair process [23]. The profile of bile acids and the community of the microbiota are in two-directional symbiosis, and various studies have shown the important effects of both in maintaining and restoring intestinal (epithelial) health.

The administration of BL1719 for 15 days resulted in a significantly (*p* = 0.031) higher concentration of aspartate in the TG1 group (median value of 31.15 µmol/L) in comparison with the CG1 group (median value of 20.85 µmol/L). At the end of the longer intervention, a somewhat higher concentration of aspartic acid was also registered in the TG2 group (median value of 33.6 µmol/L vs. median value of 28.9 µmol/L in CG 2) (Table 6). Similar results were obtained also in the case of another amino acid, glutamate, where after the 15-day intervention period, the level was significantly higher in the TG1 group compared to the CG1 group (median values of 112.5 µmol/L vs. 72.7 µmol/L., respectively, *p* = 0.026) and also remained higher during the 30-day intervention (median value in TG2 of 118 µmol/L vs. 105.5 µmol/L for CG2; non-significant difference).

Additionally, higher concentrations of the amino acids cysteine, tyrosine, and aspartic acid were observed in the serum of the mice after 15 and 30 days, as were those of arginine after the 15-day administration of BL1719 and tryptophan after the 30-day administration of BL1719 in comparison with the control group (Table 6).

Polyamines (putrescine, spermidine, and spermine) originate exogenously from dietary sources. Endogenously, polyamines are synthesised in the gut lumen by decarboxylation from the amino acids arginine and ornithine, but they also come from exfoliated enterocytes and luminal bacteria [24]. The biological impact of polyamines is targeted to cell growth and differentiation, the regulation of immune cells, the regulation of the inflammatory reaction, the enhancement of mucosal barrier, and the improvement of the integrity of the intestinal mucosa [25]. Among the polyamines, spermine, which is derived from putrescine, is the most active in various biological processes. No differences in the serum spermine levels between the CG and TG groups were observed in either exposure period. Somewhat higher spermidine and putrescine values were registered in the TG group after the 15-day and 30-day administrations of BL1719 in comparison with the respective values in the CG group (Table 6). Although the difference between the groups was not significant and despite the limitation of the experimental design that the measurement of polyamines in the faecal samples was not planned, the metabolite profiling performed in this preliminary study still opens a new research direction to elucidate the possible effect of BL1719 on amino acid and polyamine metabolism.

Although the presence of butyric acid significantly affects the health of colonocytes, the amount of various other microbial metabolites and the balance between beneficial and harmful compounds, including indole derivates and cresol compounds, also affect colon health as well as the metabolism and the systemic health of the host [26,27]. The host’s diet, in turn, has a great impact on the intestinal microbiota, influencing the amount and types of microbial metabolites.

The amino acids l-tryptophan and l-tyrosine serve as substrates for both indole derivates and cresol compounds, affecting in turn the diversity and community of the microbiota. P-cresol-SO_4_ and indoxyl-SO_4_ are produced by representatives of different genera of the intestinal obligate or facultative proteolytic anaerobes during protein breakdown and amino acid fermentation in the colon [28,29]. There are different estimates regarding the formation of these uremic toxins in human organisms. Undoubtedly, their presence in the blood over certain concentrations is considered problematic [30]. Both affect the integrity of the intestinal barrier, exert a cytotoxic effect, and are associated with dysbiosis and chronic diseases [27]. In comparison with the CG group, in the TG group fed with BL1719, lower contents of atherosclerosis risk markers and the uremic toxins p-cresol-SO_4_ and indoxyl-SO_4_ were detected at both endpoints of the experiment, i.e., on day 15 and on day 30 (Table 6).

In their study regarding the relationship between microbiota composition and the production of uremic toxins, Candeliere et al. also demonstrated strain-specific bioconversions of the uremic toxins indole and p-cresol by representatives of *Lactobacillaceae* and *Bifidobacteriaceae* [27]. Among the most promising species, the study highlighted *Bifidobacterium longum* ssp. *longum.*

Several authors have also noted the fact that the genera *Lactobacillus* and *Bifidobacterium* have a complementary effectiveness in the removal of these uremic toxins [26,27]. Both genera were represented in the top 20 genera of relative abundance in the TG1 and TG2 groups on day 15 and on day 30. Both genera were represented in the top 20 genera of relative abundance for the TG1 and TG2 groups on days 15 and 30 in comparison with the control groups’ respective indicator, where the genus *Bifidobacterium* was below the threshold and the abundance of the genus *Lactobacillus* was lower (Figure 3b).

Supplementation of the diet with dietary fibre balances the microbiota and thereby improves the composition of microbial metabolites. In the present study, the diet of the mice was not intentionally affected (e.g., high-fat or -protein diet); the experimental animals were fed with a universal commercial diet balanced with macro- and micronutrients and different fibres (wheat products, barley, oat hulls, sugar beet pulp). This, in turn, could have contributed to improving the intestinal environment.

Unlike indoxyl-SO_4_, another tryptophan-derived indole compound, 3-indolepropionic acid (3-IPA), has been reported to have possible neuroprotective, anti-inflammatory, and antioxidative effects [31]. 3-IPA was found in a somewhat higher concentration in the TG2 group compared to the CG2 group after the longer intervention (Table 6).

### 3.6. Lipid Level in Blood Serum

At the end of both intervention periods, in addition to various markers, the lipid content in the blood serum of the mice was determined. In the studied mice, the values of blood serum lipids were typical of those found in conventional mice [32,33]. As conventional mice were used in the present experiment with no intentional diet-, stress-, or otherwise-induced hypercholesterolaemia, no definitive conclusions can be drawn from this experiment regarding the effect of BL1719 on serum lipid concentrations (Table 7).

## 4. Conclusions

During the present animal study, we found that, when administered in live form, the *Bifidobacterium longum* ssp. *longum* strain BL1719 may exert health-promoting effects on the host through different mechanisms, including the modulation of the gut microbiome and the host metabolome via inducing the production of health-promoting bioactive compounds and decreasing some potentially harmful ones. It is important to confirm our preclinical findings (the results obtained in the present mouse model and during a previous in vitro study) by conducting pilot clinical trials on relatively healthy adult human volunteers to evaluate the tolerability and potential health-promoting effects of BL1719 at defined daily doses and in a postbiotic form.

## Figures and Tables

**Figure 1 microorganisms-12-00840-f001:**
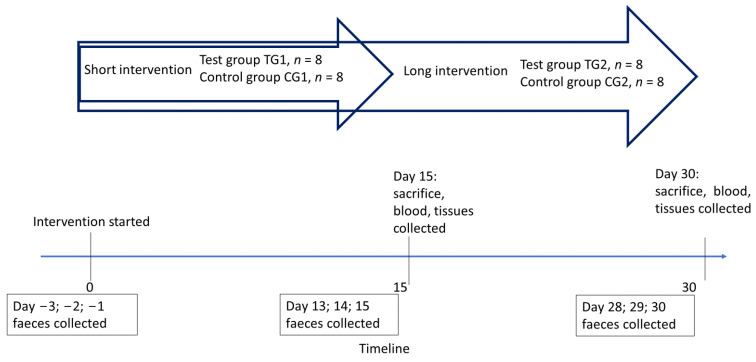
Experimental design.

**Figure 2 microorganisms-12-00840-f002:**
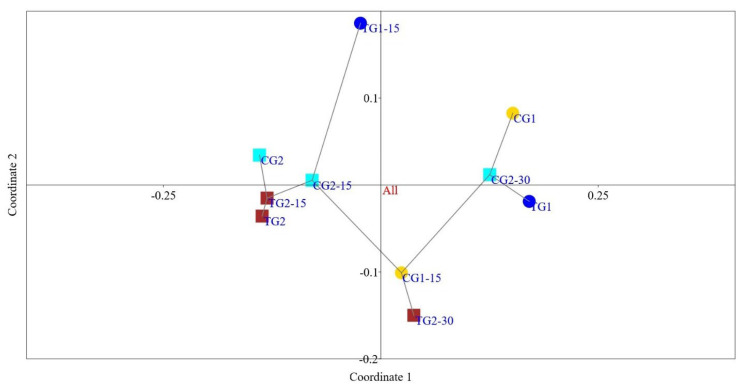
Principal coordinate analysis (PCoA) plot of bacterial communities in faecal samples based at genus level. PCoA plot demonstrates clustering of different samples per cage. Fifteen-day intervention period: CG1—control group, TG1—test group; thirty-day intervention period: TG2—test group, CG2—control group.

**Figure 3 microorganisms-12-00840-f003:**
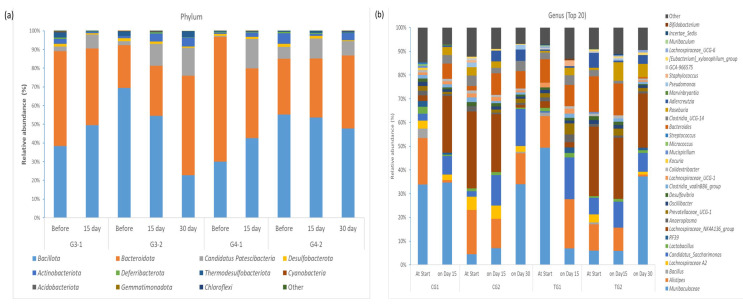
Dynamics of faecal microbial communities in mouse faeces on phylum (**a**) and genera (**b**) levels. Faecal samples were collected per cage during the three days prior to the feeding trial; CG—control group, TG—test group.

**Figure 4 microorganisms-12-00840-f004:**
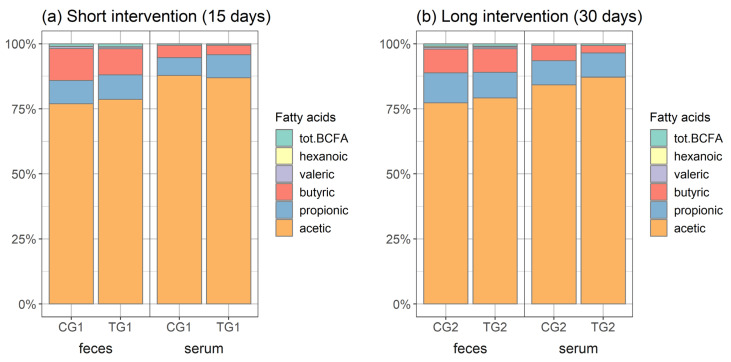
Short-chain fatty acid proportions (%) in faecal and serum samples of mice after (**a**) 15-day and (**b**) 30-day intervention periods. CG—control group, TG—test group; tot-BCFA—total branched-chain fatty acids: isobutyric acid and isovaleric acid.

**Table 1 microorganisms-12-00840-t001:** Alpha diversity indices—richness, Shannon ‘H’, and inverse Simpson (Invsimpson)—with respect to sample data (per cage).

Study Group		Richness	Shannon Diversity Index H	InvSimpson
CG1	At start	62	2.65	0.16
	On day 15	54	2.28	0.19
CG2	At start	56	2.61	0.15
	On day 15	50	2.68	0.11
	On day 30	60	2.41	0.17
TG1	At start	53	2.08	0.28
	On day 15	57	2.88	0.10
TG2	At start	53	2.54	0.14
	On day 15	55	2.66	0.12
	On day 30	57	2.30	0.20

CG—control group, TG—test group.

**Table 2 microorganisms-12-00840-t002:** Short-chain fatty acid content (µmol/g) in faecal samples of mice during the 15- and 30-day intervention periods.

	Group	Total SCFAs			Total Straight-Chain SCFAs			Total BCFAs		
		At Start	On Day 15	On Day 30	At Start	On Day 15	On Day 30	At Start	On Day 15	On Day 30
Short intervention	CG1	79.56	79.71		78.62	78.90		0.94	0.81	
	TC1	67.84	86.66		67.00	85.68		0.84	0.98	
Long intervention	CG2	89.72	92.32	89.68	88.60	91.48	88.82	1.12	0.84	0.87
	TC2	89.43	92.77	93.80	88.50	91.94	92.91	0.93	0.83	0.89

CG—control group, TG—test group; total straight-chain SCFAs: acetic acid, propionic acid, butyric acid, valeric acid, and hexanoic acid; BCFAs—branched-chain fatty acids: isobutyric acid and isovaleric acid.

**Table 3 microorganisms-12-00840-t003:** Short-chain fatty acid content in faeces (µmol/g) of mice during 15-day intervention period.

SCFA	CG1		TG1	
	At Start	On Day 15	At Start	On Day 15
Acetic acid	60.58	61.32	51.06	68.11
Propionic acid	8.17	7.15	6.68	8.17
Butyric acid	9.15	9.83	8.55	8.75
Isobutyric acid	0.40	0.36	0.37	0.45
Valeric acid	0.53	0.49	0.53	0.52
Isovaleric acid	0.54	0.45	0.46	0.54
Hexanoic acid	0.19	0.11	0.19	0.12

CG1—control group, TG1—test group.

**Table 4 microorganisms-12-00840-t004:** Short-chain fatty acid content in faeces (µmol/g) of mice during 30-day intervention period.

SCFA	CG2			TG2		
	At Start	On Day 15	Day 30	At Start	On Day 15	Day 30
Acetic acid	68.06	66.46	69.33	67.65	73.06	74.30
Propionic acid	8.78	9.48	10.35	8.27	8.33	9.25
Butyric acid	10.71	14.77	8.21	11.69	9.91	8.49
Isobutyric acid	0.49	0.37	0.39	0.41	0.37	0.39
Valeric acid	0.68	0.64	0.58	0.61	0.53	0.54
Isovaleric acid	0.63	0.47	0.48	0.52	0.47	0.50
Hexanoic acid	0.37	0.14	0.36	0.27	0.11	0.33

CG—control group, TG—test group.

**Table 5 microorganisms-12-00840-t005:** Short-chain fatty acid concentration in blood sera (µmol/L) of mice after 15-day and 30-day intervention periods.

SCFA	CG1		TG1		*p* Value CG1 vs. TG1	CG2		TG2		*p* Value CG2 vs. TG2
	On Day 15					On Day 30				
	Median	IQR	Median	IQR		Median	IQR	Median	IQR	
Acetic acid	15.89	0.55	15.37	4.00	0.27	15.23	1.91	15.14	1.60	0.54
Propionic acid	1.24	0.60	1.76	0.48	0.31	1.69	0.62	1.54	0.61	0.78
Butyric acid	0.86	0.20	0.46	0.60	0.09	1.06	0.34	0.63	0.22	0.016
Isobutyric acid	0.06	0.01	0.09	0.03	1.0	0.07	0.02	0.06	0.01	0.06
Isovaleric acid	0.04	0.01	0.03	0.02	0.06	0.03	0.01	0.02	0.02	0.60
Hexanoic acid	0.01	0.00	0.01	0.00	1.0	0.01	0.01	0.01	0.00	0.04
Lactic acid	9084	2535	9037	1637	0.965	11465	5792	9084	1966	0.129
Succinic acid	42.15	23.87	39.4	7.44	0.514	39.25	9.15	38.3	5.2	0.841

CG—control group, TG—test group; IQR—interquartile range.

**Table 6 microorganisms-12-00840-t006:** The concentrations of metabolites (µmol/L, median (IQR)) in serum of mice after 15 days and 30 days of intervention.

Bio-Chemical Class	Metabolite	CG1	TG1	*p* Value CG1 vs. TG1	CG2	TG2	*p* Value CG2 vs. TG2
		Day 15			Day 30		
Bile acids	Cholic acid (CA)	0.00 (0.00)	0.01 (0.01)	0.008	0.01 (0.02)	0.02 (0.03)	0.250
	Taurocholic acid (TCA)	0.20 (0.13)	0.25 (0.48)	0.89	0.21 (0.25)	0.17 (0.11)	0.41
	Deoxycholic acid (DCA)	0.02 (0.01)	0.030 (0.01)	0.015	0.020 (0.00)	0.050 (0.04)	0.052
Aminoacids	Tyrosine	74.45 (8.75)	93.25 (10.79)	0.186	93.20 (47.10)	103.00 (31.30)	0.575
	Tryptophan	116 (18.25)	114.5 (29.05)	0.694	104.5 (23.45)	144 (45.00)	0.320
	Aspartate	20.85 (4.60)	28.90 (12.52)	0.031	31.15 (18.18)	33.6 (14.70)	0.968
	Arginine	193.50 (36.00)	240.00 (29.25)	0.067	255.00 (68.00)	225.00 (92.00)	0.841
	Cysteine	23.95 (3.18)	26.90 (5.43)	0.056	27.20 (5.41)	30.2 (4.00)	0.071
	Glutamate	72.70 (17.80)	112.50 (18.50)	0.026	105.50 (30.90)	118.00 (27.50)	0.208
	Ornitine	117.50 (34.05)	119 (17.75)	0.769	132 (46.40)	112 (44.00)	0.904
Polyamines	Putrescine	0.85 (3.51)	0.91 (0.13)	0.760	0.80 (0.25)	0.97 (0.09)	0.089
	Spermidine	2.08 (0.59)	2.65 (0.90)	0.280	2.29 (0.97)	2.31 (1.3)	0.748
	Spermine	0.50 (0.26)	0.52 (0.37)	0.631	0.46 (0.10)	0.48 (0.24)	0.904
Cresols	p-cresol-SO_4_	1.12 (1.21)	1.09 (1.69)	0.460	0.49 (0.72)	0.14 (0.20)	0.073
Indoles and derivatives	Indoxyl-SO_4_	14.72 (19.01)	10.14 (18.08)	0.514	9.39 (6.89)	5.33 (6.20)	0.186
	3-indolepropionic acid	0.59 (0.72)	0.62 (0.91)	1.00	0.46 (0.62)	0.73 (1.13)	0.061

CG—control group, TG—test group; IQR—interquartile range.

**Table 7 microorganisms-12-00840-t007:** Lipid concentration (mmol/L) in blood serum of mice after 15 days and 30 days of intervention.

Variable	Day 15				*p* Value CG1 vs. TG1	Day 30				*p* Value CG2 vs. TG2
	CG1		TG1			CG2		TG2		
	Median	IQR	Median	IQR		Median	IQR	Median	IQR	
Total cholesterol	2.87	0.19	2.75	0.3	0.689	2.63	0.2	2.56	0.62	1
LDL cholesterol	0.12	0.01	0.13	0.04	0.069	0.13	0.06	0.15	0.06	0.411
HDL cholesterol	2.41	0.18	2.36	0.19	0.476	2.25	0.18	2.02	0.51	0.902
Triglycerides	1.96	0.31	1.39	0.8	0.198	1.58	0.37	1.95	1.09	0.438

CG—control group, TG—test group; IQR—interquartile range.

## Data Availability

Data are contained within the article.

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
