# Peer review of "Investigation of Effects of Novel Bifidobacterium longum ssp. longum on Gastrointestinal Microbiota and Blood Serum Parameters in a Conventional Mouse Model"

_microorganisms, 2024, doi:10.3390/microorganisms12040840_

Round 1

Reviewer 1 Report

Comments and Suggestions for Authors

Research into the biological, biotechnological, and safety properties of probiotic strains is currently relevant. In this context, studies on model animals conducted by the authors are essential. The introduction contains the necessary and relevant information to justify the purpose of the study.

The design and methodological sections of the manuscript require additional explanation. The main comments are caused by the description of the results.

1. L. 45: Provide the correct definition of the abbreviation HMOs.

2. L. 96-97 states that 16 mice were included in the study. However, when describing the division into groups and subgroups, it is said that in each group (L. 104-105) "resulting in a total of four groups of mice: TG1, TG2, CG1, and CG2, each 8 animals per cage (Figure 1)", which in total is 32 mice. The design needs to be clarified.

3. L. 125. It is necessary to clarify at which points tissue samples were taken for histological analysis.

It is necessary to include how and at what points blood samples were collected.

4. L. 133-137: It is necessary to clarify how the cultivation results were considered.

5. L. 220: It’s better to clarify – 30 days.

6. L. 232: clarify the ranks for which the results “including 27 genera, 60 genera” are given.

7. Figure 1 indicates that the fecal biomaterial was collected at three points sequentially over three days. Therefore, it is necessary to clarify the data presented in tables 1, 2, and 3: at start – what day is it -1, -2, or -3? Why is the analysis not provided for days 13 and 14, 28 and 29? If the table presents average values, this must be indicated.

Additionally, you should enter static values as confirmation of whether it is significant or not. Describing the result as more vs. less is not acceptable in this context.

8. Taxonomy (especially at the phylum level) throughout the text needs to be brought into line with modern taxonomy. There is no genus Bifidobacteria, correctly Bifidobacterium.

Check the spelling of the species and generic names of bacteria in italics.

9. It is not clear why p values are not calculated in tables 1.3-5, but they are presented in tables 6 and 7. It is necessary to add p values to tables 1, 3-5 so that the presentation of the results is more accurate.

In general, after all corrections have been made, the work can be published.

Reviewer 2 Report

Comments and Suggestions for Authors

In this study the authors “investigation of effects of novel Bifidobacterium longum ssp.  longum on gastrointestinal microbiota and blood serum parameters in a conventional mouse model, and found that strain BL1719 may exert health-promoting effects on the host through different mechanisms, including the modulation of gut microbiome and the host metabolome via inducing the production of health-promoting bioactive compounds and decreasing some potentially harmful ones.”

Comments:
1. Please add a detail about the analysis of the total cholesterol, triglycerides, HDL-cholesterol and LDL-cholesterol in the section of materials and methods 2.2.7.
2. “Bile acids are involved in lipid metabolism including cholesterol biosynthesis and elimination, the promotion of lipid and fat-soluble vitamins absorption, etc.
Primary bile acids are synthesized de novo by the liver from cholesterol. Secondary bile acids are produced by intestinal microbes from primary bile acids which are excreted 413 into the intestinal tract as glycine or taurine conjugates“ please add the references.

Round 2

Reviewer 1 Report

Comments and Suggestions for Authors

The authors have made almost all corrections to the text, except for comment 8. Considering the specificity of the journal, these comments should be considered by the authors in full.

I repeat that taxonomy, especially the spelling of phyla, should be given according to modern taxonomy; see https://www.bacterio.net/. The authors should revise the names of all taxa (mainly phyla names) described on pages 8-12, including Table 2 and Figure 3.

In addition, the text should be carefully proofread and сheck the spelling of the species and genera names of bacteria should be changed in italics.

For example, for the phyla Firmicutes and Patescibacteria, the correct spelling is Bacillota and Candidatus Patescibacteria. 

Name: "Firmicutes" (Gibbons and Murray 1978) Garrity and Holt 2001

Nomenclatural status: not validly published

Taxonomic status: synonym (and no standing)

Correct name: Bacillota Gibbons and Murray 2021

Name: "Candidatus Patescibacteria" Parks et al. 2018

Category: Phylum

Proposed as: Candidatus
